Biogeography rather than substrate type determines bacterial colonization dynamics of marine plastics

Coons Ashley K. 1
Busch Kathrin 1
Lenz Mark 1
Hentschel Ute 1 2
Borchert Erik eborchert@geomar.de 1
1 GEOMAR Helmholtz Centre for Ocean Research Kiel , Kiel , Schleswig-Holstein , Germany
2 Christian-Albrechts-University Kiel , Kiel , Schleswig-Holstein , Germany
Brazelton William
Electronic publication date: 2021 Sep 13
Publication date: 2021
Volume: 9
Electronic Location ID: e12135
Received 2021 Apr 22; Accepted 2021 Aug 18
Copyright: ©2021 Coons et al.
Copyright year: 2021
Copyright holder: Coons et al.
License: This is an open access article distributed under the terms of the Creative Commons Attribution License, which permits unrestricted use, distribution, reproduction and adaptation in any medium and for any purpose provided that it is properly attributed. For attribution, the original author(s), title, publication source (PeerJ) and either DOI or URL of the article must be cited.
License URL: https://creativecommons.org/licenses/by/4.0/

Keywords: Marine plastics, Plastisphere, Bacterial colonization

Funding: PLASTISEA as part of the BMBF 031B0867A DFG This work was supported by PLASTISEA as part of the BMBF Funding Activity ‘New Biotechnological Processes based on Marine Resources—BioProMare’ 2020–2023 (Funding Reference Number: 031B0867A) (awarded to Ute Hentschel and Erik Borchert). We received financial support from DFG within the funding program Open Access Publizieren. The funders had no role in study design, data collection and analysis, decision to publish, or preparation of the manuscript.

==============================
Since the middle of the 20th century, plastics have been incorporated into our everyday lives at an exponential rate. In recent years, the negative impacts of plastics, especially as environmental pollutants, have become evident. Marine plastic debris represents a relatively new and increasingly abundant substrate for colonization by microbial organisms, although the full functional potential of these organisms is yet to be uncovered. In the present study, we investigated plastic type and incubation location as drivers of marine bacterial community structure development on plastics, i.e., the Plastisphere, via 16S rRNA amplicon analysis. Four distinct plastic types: high-density polyethylene (HDPE), linear low-density polyethylene (LDPE), polyamide (PA), polymethyl methacrylate (PMMA), and glass-slide controls were incubated for five weeks in the coastal waters of four different biogeographic locations (Cape Verde, Chile, Japan, South Africa) during July and August of 2019. The primary driver of the coastal Plastisphere composition was identified as incubation location, i.e., biogeography, while substrate type did not have a significant effect on bacterial community composition. The bacterial communities were consistently dominated by the classes Alphaproteobacteria, Gammaproteobacteria, and Bacteroidia, irrespective of sampling location or substrate type, however a core bacterial Plastisphere community was not observable at lower taxonomic levels. Overall, this study sheds light on the question of whether bacterial communities on plastic debris are shaped by the physicochemical properties of the substrate they grow on or by the marine environment in which the plastics are immersed. This study enhances the current understanding of biogeographic variability in the Plastisphere by including biofilms from plastics incubated in the previously uncharted Southern Hemisphere.

Introduction

Since the initial mass-production of plastics as inexpensive, single-use, sanitary health and convenience items during the 1950s, these synthetic polymers have been rapidly integrated into nearly every aspect of our daily lives. As of 2019, annual global plastic production exceeded 365 million metric tons (MT) (PlasticsEurope, 2020) and is widely recognized as a pollutant in virtually all environments—both terrestrial and aquatic, freshwater and marine (De Souza Machado et al., 2018; Harrison et al., 2018). It has been estimated that 3% of plastic produced annually enters the ocean each year, largely via riverine input, municipal wastewater effluent, and litter produced by urban tourism (Jambeck et al., 2015; Auta, Emenike & Fauziah, 2017). Plastic debris often concentrates in oceanic gyres (Law et al., 2010), but has also been discovered in remote regions, including Arctic Sea ice (Peeken et al., 2018), and at depths greater than 4,000 m in the Pacific Ocean (Krause et al., 2020). At first, the attention of the media and scientists focused on the more apparent negative effects of larger plastic debris, including entanglement and ingestion, in the marine environment (Laist, 1997). Throughout the past decade, research efforts have shifted towards the ecological impacts of microplastics (<5 mm diameter; Arthur, Baker & Bamford, 2009) on marine animals such as bivalves (Sussarellu et al., 2016), fish (Lusher et al., 2016), and zooplankton (Cole et al., 2013), and the inevitable link to humans through food web interactions (Cox et al., 2019). Most recently, concurrent with swift advancements in molecular techniques, researchers have begun to describe the microbial life colonizing marine plastic debris in an effort to clarify which microorganisms are present (Zettler, Mincer & Amaral-Zettler, 2013).

The moment debris comes into contact with the environment, a biofilm, defined as a community of microorganisms that is attached to a surface (O’Toole, Kaplan & Kolter, 2000), begins to form (Dang & Lovell, 2000). The term “Plastisphere” was coined by Zettler, Mincer & Amaral-Zettler (2013) to describe the biofilm-forming communities on the surfaces of marine plastic debris—an immediate process that encompasses different domains of life (Amaral-Zettler, Zettler & Mincer, 2020). Most Plastisphere research has highlighted findings of prokaryotic, especially bacterial diversity (Wright, Langille & Walker, 2020), although some reports of eukaryotic (Bryant et al., 2016; Zaiko et al., 2016; Kettner et al., 2019; Oberbeckmann et al., 2014), including fungal (Gonda, Jendrossek & Molitoris, 2000; Kettner et al., 2017; Lacerda et al., 2020), diversity of mature biofilms on marine plastic debris have been published.

Previous research investigating the bacterial Plastisphere has considered a variety of variables that can potentially influence the formation and resulting composition of the biofilm community, including substrate type (Ogonowski et al., 2018; Kirstein et al., 2019; Oberbeckmann & Labrenz, 2020; Wright et al., 2020), biogeography (Oberbeckmann et al., 2014; Amaral-Zettler et al., 2015), seasonality (Oberbeckmann et al., 2014), and age of biofilm (Harrison et al., 2014; Wright, Langille & Walker, 2020). Many distinct synthetic polymer types, primarily polyethylene (PE), polypropylene (PP), polystyrene (PS), and polyethylene terephthalate (PET) (supplementary information; Wright, Langille & Walker, 2020), have been utilized as polymer substrates to investigate the ecology of the Plastisphere. Marine plastic debris is dominated by PP and PE (Erni-Cassola et al., 2019), the plastic types of the highest consumer demand (PlasticsEurope, 2020). Current knowledge regarding the influence of biogeographic location on microbial Plastisphere communities is focused on the Northern Hemisphere (Wright et al., 2020), while the Southern Hemisphere is so far scientifically underrepresented (Wright, Langille & Walker, 2020). Various experiments in coastal- and open-ocean surface waters have been conducted, including the Pacific (Bryant et al., 2016; Zaiko et al., 2016; Tobias-Huenefeldt et al., 2021), Atlantic (Debroas, Mone & Ter Halle, 2017) and Indian Ocean (Muthukrishnan, Al Khaburi & Abed, 2019), as well as the Baltic (Oberbeckmann et al., 2014; Oberbeckmann, Kreikemeyer & Labrenz, 2018; Kesy et al., 2019), North (Oberbeckmann, Osborn & Duhaime, 2016), and Mediterranean Sea (Dussud et al., 2018). Current research agrees that Plastisphere communities differ significantly from the microbial assemblages found in the surrounding seawater (Amaral-Zettler, Zettler & Mincer, 2020) and those present on naturally-occurring substrates, such as wood (Zettler, Mincer & Amaral-Zettler, 2013). Certain bacterial phyla, such as Bacteroidota and Proteobacteria (Roager & Sonnenschein, 2019), recur across geographic locations, including species of the potentially pathogenic Vibrio genus (Zettler, Mincer & Amaral-Zettler, 2013; Kirstein et al., 2016; Debroas, Mone & Ter Halle, 2017).

The aim of the present study was to determine whether, and to what extent, incubation location or plastic type influence the bacterial community composition of marine plastic biofilms. Plastic-coated glass slides and plastic-free glass-slide controls were incubated for five weeks in the upper one meter of the coastal water column in four distinct marine locations (Cape Verde, Chile, Japan, South Africa; see Fig. 1). Subsequently, 16S rRNA gene amplicon data (V3-V4 region) were generated to obtain insights into the bacterial colonization patterns on various plastic types at different locations. The results of this study expand our current understanding of taxonomic variability in microbial Plastisphere communities by including the marine biofilms originating from plastics incubated in the uncharted Southern Hemisphere.

Figure 1 Experimental setup and incubation locations.

In this study, bacterial biofilm communities that developed on various substrate types were compared between four incubation locations (A) Cape Verde (Mindelo; 16°53′11.2″N, 24°59′27.6″W), Chile (Coquimbo; 29°57′59.5″S, 71°21′.11.6″W), Japan (Akkeshi; 43°01′14.3″N, 144°50′12.0″E), and South Africa (Cape Town; 33°54′.35.89″S, 18°25′10.15″E). At each location, four plastic types (HDPE, LDPE, PA, PMMA) were melted onto individual glass microscopy slides, which were then incubated in duplicate for five weeks in the upper one meter of the coastal water column along with plastic-free glass-slide controls. (B) Bacterial biofilm community structure was assessed via amplicon analysis of the V3–V4 region of the 16S rRNA gene.

Materials and Methods

Sample collection

Plastic-coated glass slides were incubated at four sites globally by participants of the international research and student training program GAME (Global Approach by Modular Experiments) following a standardized protocol in July and August 2019. Incubation locations included Cape Verde, Chile, Japan, and South Africa (for exact locations see Fig. 1A) Environmental parameters and sampling site characteristics can be found in the Table S1. Before marine exposure, one of four distinct synthetic polymer types, i.e., polyamide (PA; PandaParticles UG, Erfurt, Germany), high-density polyethylene (HDPE; ExxonMobil), linear low-density polyethylene (LDPE; ExxonMobil, Hamburg, Germany), polymethyl methacrylate (PMMA; Kunststoff und Farben GmbH, Biebesheim, Germany), were individually melted onto one side of glass microscope slides (Superfrost® Plus; Menzel GmbH, Braunschweig, Germany) (Fig. 1B). The four different polymers were chosen based on their classification as thermoplastics and therefore no chemical alteration due to the melting process was expected. Furthermore, the four polymers can be distinguished by their buoyancy with HDPE (density of 0.941 g/cm3) and LDPE (0.915–0.94 g/cm3) being positively buoyant and PA (1.14 g/cm3) and PMMA (1.18g/cm3) being negatively buoyant. At each incubation location, ten microscope slides, i.e., two slides of each polymer type plus two uncoated glass slides as controls, were deployed vertically and at approximately one-meter depth in the coastal water column for exactly five weeks, each individually secured with fishing line and an eight-gram weight to guarantee vertical positioning. Upon sample collection, the slides were preserved in a stabilization solution (25 mM sodium citrate, 20 mM EDTA, 70 g ammonium sulfate per 100 ml, pH 5.2) and sent to GEOMAR for subsequent DNA extraction. All genetic material was acquired according to the Nagoya Protocol (Buck & Hamilton, 2011) and the Convention on Biological Diversity’s (CBD) access and benefit-sharing (ABS) regulations (European Union, No 511/ 2014). Field experiments were approved by Ministerio da Agricultura e Ambiente, Cabo Verde (approval number 012/DNA/2021); Biodiversity and coastal research department of environmental affairs, South Africa (approval number RES2019/96); Office for Mainstreaming Biodiversity, Biodiversity Policy Division, Nature Conservation Bureau, Ministry of the Environment, Japan; and Profesional Departamento de Conservación de Especies División de Recursos Naturales y Biodiversidad, Ministerio del Medio Ambiente, Gobierno de Chile.

Extraction of nucleic acids

The mature biofilms, which accumulated on the incubated slides, were chemically digested using an alkaline lysis buffer during a heat treatment (alkaline-lysis method; Kennedy et al., 2008). In brief, the biological material scraped from the surface of each slide was mixed with pre-heated lysis buffer (100 mM Tris-HCl, 100 mM EDTA, 1.5 M NaCl, 1% CTAB, 2% SDS, pH 8.0) and incubated in a water bath for two hours at 70 °C with occasional mixing. The suspension was subsequently centrifuged at 4 °C at 13,000× g for 30 min, and the clear supernatant was transferred to a fresh tube and mixed with 0.7 ×volume isopropanol. After at least 30 min of incubation at room temperature, the mixture was centrifuged again and the resulting DNA pellet was washed with 70% ethanol (≥99.8% denatured ethanol; Carl Roth®, Karlsruhe, Germany), centrifuged, and air-dried before it was resuspended in a suitable amount of Tris-EDTA (TE) buffer (10 mM Tris–HCl, 1 mM EDTA). The quality and quantity of the extraction was evaluated with a NanoDrop spectrophotometer (Desjardins & Conklin, 2010). Using the 16S rRNA gene primer pair 27F and 1492R, segments of the resulting DNA extracts were amplified for a quality check via polymerase chain reaction (PCR), with the following PCR conditions: an initial denaturation at 95 °C for 3 min, 34 cycles of 95 °C for 30 s, 56 °C for 30 s, and 72 °C for 90 s, followed by a final extension at 72 °C for 5 min and held at 10 °C. The resulting PCR products were visually assessed via 1% gel electrophoresis. For amplicon sequencing, the V3-V4 hypervariable region of the 16S rRNA gene was amplified using primer pair 341F (5′-CCTACGGGAGGCAGCAG-3′; Muyzer, De Waal & Uitterlinden, 1993) and 806R (5′-GGACTACHVGGGTWTCTAAT-3′; Caporaso et al., 2011) with the cycler conditions as follows: initial denaturation at 98 °C for 30 s, 30 cycles of 98 °C for 9 s, 55 °C for 60 s, and 72 °C for 90 s, followed by a final extension at 72 °C for 10 min and held at 10 °C. Sequencing of the V3-V4 region of the 16S rRNA gene was conducted using v3 chemistry on a MiSeq Illumina sequencing platform at the Competence Centre for Genomic Analysis (CCGA) Kiel, Germany.

Quantitative Insights into Microbial Ecology (QIIME2) pipeline

Raw amplicon sequences were processed using the open-source Quantitative Insights into Microbial Ecology (QIIME2) framework (version 2019.10; Bolyen et al., 2019) similar to the procedures described by Busch et al. (2021). For this, forward primers and heterogeneity spacers were trimmed from forward-only single-end fastq files using the cutadapt plugin (Martin, 2011). The quality of the demultiplexed reads was verified using the quality-filter plugin for PHRED-based filtering and trimming (Bokulich et al., 2013). Reads were denoised using the denoise-single method of the DADA2 algorithm (Callahan et al., 2016), which truncated the 3′ ends at 270 base pairs, removed chimeric sequences, and inferred sample composition using a parametric error model. Truncation at 270 nt length increased the quality of the reads significantly, but reduced the overlap between forward and reverse reads and therefore only forward reads were used for the analysis.

Amplicon sequence variant (ASV; Callahan, McMurdie & Holmes, 2017) taxonomy was classified at an 80% confidence level using the most recent SILVA 138 16S rRNA gene reference database (Quast et al., 2013; Yilmaz et al., 2014) via the pre-fitted classify-sklearn taxonomy classifier method (Pedregosa et al., 2012) of the feature-classifier plugin (Bokulich et al., 2018). Common eukaryotic contaminants (chloroplasts, mitochondria) and unassigned sequences were removed using the filter-features method of the feature-table plugin, then the filtered dataset was rarefied to 8,000 sequences due to a satisfactory saturation of the alpha rarefaction curves for this number of features (see Fig. S1). A phylogenetic backbone tree was constructed using both FastTree (Price, Dehal & Arkin, 2009; Price, Dehal & Arkin, 2010) and MAFFT (Katoh & Standley, 2013) alignment via the phylogeny plugin, and the resulting tree was used to compute core diversity metrics. QIIME2 artifacts containing phylogenetic and non-phylogenetic diversity metrics were computed for downstream analyses along with an alpha-rarefaction curve via the diversity plugin. QIIME2 scripts can be found in File S1.

Diversity measures

Further statistical analyses were computed using the community ecology package vegan (version 2.5-6; Oksanen et al., 2010; Oksanen et al., 2019) and stats (version 3.6.2) within the open-source R environment (version 3.6.2; R Core Team, 2019) using RStudio (version 1.1.453; RStudio Team, 2016), then graphically visualized with the aid of ggplot2 (version 3.3.0; Wickham, 2016) and ggpubr (version 0.2.5; Kassambara, 2020). Some figures were further manipulated using the open-source vector graphic editor Inkscape™ (version 0.94.4; Inkscape Project, 2019). The alpha diversity within each group of the rarefied dataset was determined by the evenness (Pielou, 1966) and phylogenetic diversity (Faith’s PD; Faith & Baker, 2006). Non-phylogenetic (evenness) and phylogenetic (Faith’s PD) diversity indices were visualized in violin plots to assess alpha diversity when replicates were grouped by substrate type and location. All replicates were included to compare between substrate types, although control replicates, i.e., communities from plastic-free glass slides, were removed before analyzing the influence of location on Plastisphere communities. The non-parametric Kruskal-Wallis rank sum test (Kruskal & Wallis, 1952; Hollander & Wolfe, 1973; Mcdonald, 2014) was implemented to determine whether the medians of the sample types differed significantly. If a significant result was observed, a Wilcoxon pairwise comparison test (Mann & Whitney, 1947) was performed with Benjamini & Hochberg correction (1995) to discover which sample types were different.

The data were explored for factors driving microbial community composition between sample types with the assistance of the R package phyloseq (version 1.30.0; McMurdie & Holmes, 2013). The qiime2R (version 0.99.21; Bisanz, 2018) package allowed for import of QIIME2 artifacts into R for the creation of a phyloseq object. Absolute count data were transformed into compositional data with the pseq.rel function, then an ordination was performed on the transformed phyloseq object using the non-metric multidimensional scaling method (NMDS; Kruskal, 1964) with a sample-wise unweighted UniFrac distance matrix (Lozupone & Knight, 2005). The visual interpretation of the NMDS plot was confirmed with a non-parametric, permutational multivariate analysis of variance (PERMANOVA; Anderson, 2001; Anderson, 2017) test. The PERMANOVA group significance and pairwise tests were run simultaneously via the beta-group-significance method (non-parametric MANOVA; Anderson, 2001) of the QIIME2 diversity plugin with an unweighted UniFrac matrix and 999 permutations. A significance level of α = 0.05 was applied for all statistical analyses.

Taxonomic composition analyses

Bubble plots were used to display phyla that represented more than 0.01% of the medians of the relative community composition per substrate type per incubation location. Furthermore, a sunburst diagram was created to show the median relative composition of the families that made up the most abundant bacterial phyla on plastic replicates. Taxonomic assignments on family level responsible for less than 0.5% of the median relative abundance and order level with a cumulative median relative abundance of less than 1% were removed and were then combined at a higher taxonomic level. The web-based tool InteractiVenn (Heberle et al., 2015) produced a Venn diagram to display ASVs distinct to or shared between plastic replicates from each incubation location. The rarefied ASV table produced in QIIME2 was converted into a binary presence-absence table using the feature-table plugin. The resulting biom table was exported and converted to a tab-delimited file using the biom convert command, then ASVs with null values or present in glass-slide controls were excluded.

Results

Sample overview

In total, 36 samples from the four coastal incubation locations: Cape Verde (n = 10); Chile (n = 8); Japan (n = 8); South Africa (n = 10) met quality control and minimum library size requirements. The QIIME2 pipeline was completed with these 36 samples consisting of four polymer types: HDPE (n = 7), LDPE (n = 7), PA (n = 8), PMMA (n = 7), and glass-slide controls (n = 7). Positive and negative sequencing controls were also removed. After quality control, truncation (270 bp), removal of eukaryotic contaminants and unassigned reads, and subsampling to the lowest number of reads (alpha rarefaction curve sufficiently saturated at 8,000 features; Fig. S1), 721,330 non-chimeric sequences remained from the initial 949,349 demultiplexed Illumina reads with an average sequence frequency of 20,036 reads across the 36 samples. The reads were made up of 12,361 unique features, which were taxonomically assigned according to the SILVA single subunit (SSU) database release 138 (80% confidence) (Quast et al., 2013; Yilmaz et al., 2014). The observed ASVs per sample ranged from 403 to 1,353 sequences. The ASV and taxonomy table can be found in the Tables S2 and S3.

Diversity measures

The Shannon diversity indices ranged from 7.6 to 9.7 per sample, Pielou’s evenness from 0.57 to 0.92 and Faith’s PD from 35.1 to 98.3. Evenness (Kruskal-Wallis rank sum test: chi-squared = 5.089, p = 0.278, df = 4) and cumulative phylogenetic diversity (chi-squared = 2.179, p = 0.703, df = 4) of the bacterial communities did not vary significantly by substrate type (Fig. 2). Pielou’s evenness (chi-squared = 16.663, p < 0.001, df = 3) and Faith’s PD (chi-squared = 13.649, p = 0.003, df = 3) did, however, vary significantly by location (Fig. 3). Pairwise Wilcoxon rank sum tests revealed a significantly higher evenness of bacterial communities on plastic replicates from Cape Verde, when compared to those from mid-latitudes (Chile, Japan, and South Africa) (pairwise Wilcoxon tests: p < 0.001). Communities from Chile exhibited significantly lower phylogenetic diversity than those from Cape Verde (pairwise Wilcoxon test: p = 0.028), Japan (p = 0.007), and South Africa (p = 0.018). Communities from the two Northern Hemisphere locations were not significantly different from one another phylogenetically (Cape Verde, Japan; p = 1.0), although they both displayed a significantly higher diversity than the assemblages from the two Southern Hemisphere locations (p = 0.001; Fig. S2).

Figure 2 Influence of plastic type on alpha diversity.

Violin plots displaying (A) Pielou’s evenness (Kruskal–Wallis rank sum test: chi-squared = 5.089, p = 0.278, df = 4) and (B) Faith’s PD (chi-squared = 2.179, p = 0.703, df = 4) within grouped replicates of each substrate type (N = 36): glass-slide control (n = 7), HDPE (n = 7), LDPE (n = 7), PA (n = 7), and PMMA (n = 8). Non-significant p-values are not shown.

Figure 3 Influence of incubation location on alpha diversity.

Violin plots displaying (A) Pielou’s evenness (Kruskal–Wallis: chi-squared = 16.663, p < 0.001, df = 3) and (B) Faith’s PD (chi-squared = 13.649, p = 0.003, df = 3) of grouped plastic replicates within each study location (N = 29): Cape Verde (n = 8), Chile (n = 7), Japan (n = 6), South Africa (n = 8). *, **, and *** represent p-values ≤ 0.05, ≤ 0.01, and ≤ 0.001, respectively. Non-significant p-values are not shown.

Bacterial Plastisphere communities clustered most clearly by location when visualized via NMDS ordination (Fig. 4). A PERMANOVA test confirmed that location had a significant influence on community composition (PERMANOVA: pseudo-F = 7.516, p = 0.001; Table 1A), while substrate type did not (PERMANOVA: pseudo-F = 0.637, p = 0.999; Table 1A). Pairwise, non-parametric MANOVA tests revealed significant differences between all incubation locations (MANOVA results; Table 1B).

Figure 4 Incubation location is a primary driver of bacterial community composition.

Nonmetric multidimensional scaling (NMDS) ordination based on an unweighted UniFrac distance matrix of compositionally transformed microbial abundances. Each location (Cape Verde, Chile, Japan, South Africa) is represented by a distinct color and each substrate type (glass-slide control, HDPE, LDPE, PA, PMMA) by a unique symbol. Run stress value: 0.081.

Table 1 PERMANOVA results.

(A) Results of the group-wise PERMANOVA statistical test for substrate type, location, and origin parameters. Pairwise MANOVA tests were completed by location (B), as they had more than two comparable groups.

(A)						
Parameter	Sample size	Permutations	Pseudo-F	p-value	Groups	
Substrate type	36	999	0.637	0.999	5	
Location	29	999	7.516	0.001	4	
Hemisphere	29	999	6.082	0.001	2	
(B)							
Location A	Location B	Sample size	Permutationsss	Pseudo-F	p-value	q-value	
Cape Verde	Chile	15	999	9.337	0.001	0.001	
Cape Verde	Japan	14	999	7.354	0.001	0.001	
Cape Verde	South Africa	16	999	7.745	0.001	0.001	
Chile	Japan	13	999	7.179	0.003	0.003	
Chile	South Africa	15	999	6.247	0.001	0.001	
Japan	South Africa	14	999	6.940	0.001	0.001	

Taxonomic composition analyses

The 12361 unique ASVs represented 49 bacterial phyla, and only 0.34% of these were not classified beyond the kingdom level (Fig. 5). Proteobacteria were the most abundant phylum, followed by Bacteroidota. Combined, the two phyla accounted for more than 65% of the bacterial community of each sample, irrespective of substrate type or location (Fig. 5). Eleven phyla accounted for >93.1% of each community, in descending order: Proteobacteria (46%–81.2%), Bacteroidota (6.8%–35.8%), Verrucomicrobiota (0.7%–12.1%), Bdellovibrionota (0.3%–3.5%), Actinobacteriota (<0.1%–8.9%), Planctomycetota (<0.1%–3.6%), Patescibacteria (<0.1%–3.7%), Desulfobacterota (0.2%–4.2%), Acidobacteriota (<0.1%–7.1%), Myxococcota (0.1%–3%), and Cyanobacteria (<0.1%–7.2%). Differences in the identity of the most abundant phyla were more pronounced between incubation locations than between substrate types. The remaining 38 phyla accounted for less than 7% of the total bacterial diversity per sample.

Figure 5 Phylum level composition of the bacterial communities.

Proteobacteria and Bacteroidota account for the largest proportion of bacterial communities across all samples irrespective of substrate type or incubation location. The bubble plot displays the relative abundance (depicted by size) of phyla that represent more than 0.01% of the bacterial community structure on at least one substrate type (HDPE, LDPE, PA, PMMA, glass-slide control) in at least one location (Cape Verde, Chile, Japan, South Africa). Phyla are listed in descending order, replicates are faceted by location, and median values are displayed for substrate types with more than one replicate in each location. *Sample types without replicates.

Substrate type was not a significant driver of bacterial community composition (PERMANOVA, p = 0.999; Table 1A). For this reason, glass-slide controls were removed from the analysis to allow for family-level evaluation across all plastic replicates. Proteobacteria and Bacteroidota were fully represented by the classes Alpha- and Gammaproteobacteria (14.9% and 25.1% of the median relative abundance, respectively), Bacteroidia (19.5%), and a very small percentage of Rhodothermia (0.1%). Families accounting for >0.5% of the median relative community structure on plastics had a high level of overlap with previous studies on bacterial Plastisphere communities (Fig. 6) (Amaral-Zettler, Zettler & Mincer, 2020).

Figure 6 Median percental abundances of recurring bacterial families in Plastisphere communities.

Sunburst chart displaying the median relative abundance of families belonging to the highly prevalent bacterial classes according to the total obtained read counts (447,415 reads) from all plastic samples: Gammaproteobacteria, Bacteroidia and Alphaproteobacteria. Median percental abundances are indicated for all families that reached values above 1.5%. Family names are displayed for taxa that account for more than 0.5% of the median relative abundance across all plastic replicates. Order names were removed when the overall median abundance was less than 1% of the total median community composition.

ASVs found on glass slide controls were further removed to create an ASV-level comparison of the bacterial communities present on plastic replicates between sampling locations (Fig. 7). The resulting 7825 ASVs present on the plastic polymer-coated slides were principally determined by location. Japan accounted for the largest proportion of distinct ASVs (32.0%), followed by Cape Verde (29.4%), South Africa (20.6%), and Chile (14.1%). An overlap in ASVs was not observed between all four incubation locations. Excluding the equatorial Cape Verde location, the three mid-latitude sites have 18 ASVs in common. Two of these ASVs were classified to species level, and represent Ilumatobacter nonamiensis strain YM16-303 (Actinobacteriota; 99.4% confidence) and Portibacter lacus (Bacteroidota; 99.9%). The remaining 16 ASVs were classified to varying degrees, belonging mostly to Bacteroidota and Proteobacteria. The ASVs belonging to Bacteroidota were classified to genus (Portibacter, 99.7%; Marinoscillum, 91.5%; Reichenbachiella, 99.9%; Kordia, 99.9%) or family level (Cryomorphaceae, 99.9%; Flavobacteriaceae, 99.9%). The ASVs that belonged to Proteobacteria were classified to genus (Kordiimonas 99.9%; Altererythrobacter, 80.9%; Pseudoalteromonas, 98.4%; Woeseia, 99.5%), order (Xanthomonadales, 86.5%), and class level (Gammaproteobacteria, 99.9%). The four remaining ASVs were classified to genus level: SM1A02 (Planctomycetota; 98.3%), R76-B128 (Verrucomicrobiota; 99.9%), and Haloferula (Verrucomicrobiota; 96.3%), plus one order-level classification: Bradymonadales (Desulfobacterota; 95.0%). These four ASVs could be further identified by NCBI BLAST to be related to the genera Phycisphaera, Kiritimatiella, Haloferula and Nitrospina, respectively (see Table S4 for all BLAST results). Furthermore, each location was investigated for ASV distribution among the different plastic types excluding the ASVs also found on the glass control slides (Fig. S3). At each location several unique ASVs were found per plastic type.

Figure 7 Shared and distinct ASV counts by incubation location.

Venn diagram depicting distinct and shared ASV counts (N = 7825) of pooled-polymer substrates per location: Cape Verde (29.4% distinct), Chile (14.06%), Japan (32.0%), South Africa (20.6%). ASVs found in biofilms of control samples were removed. This diagram was created using InteractiVenn.net.

Discussion

Influence of incubation location on bacterial community composition

The coordinated incubation of plastic-coated glass slides allowed for a comparison of marine bacterial Plastisphere communities between four distinct incubation locations (Cape Verde, Chile, Japan, South Africa) distributed between Earth’s two largest oceans (Atlantic and Pacific) and in both hemispheres. Well-documented research regarding marine diversity in surface waters has revealed high diversity near continental margins, which decreases longitudinally towards open-ocean environments (Gray, 1997) and latitudinally towards the poles for microbial organisms (Martiny et al., 2006; Fuhrman et al., 2008; Tara Oceans: Ibarbalz et al., 2019; Salazar et al., 2019). Parallel trends were reflected in this study, suggesting a larger influence of the biogeographic location than substrate type on bacterial Plastisphere community structure. When averaged across all polymer types, plastic-coated glass slides from all locations were similar in evenness (Fig. 2), while, when averaged across all substrate types, the communities from the equator (Cape Verde) had a significantly higher evenness than the mid-latitude incubation locations (Chile, Japan, South Africa) (Fig. 3). Here, temporal variation in environmental variables is characteristically less pronounced than in temperate or polar regions (Bunse & Pinhassi, 2017). In contrast to this, the phylogenetic diversity of the communities that established on the plastic-coated slides varied between locations and was significantly lower on slides from the Southern Hemisphere (Fig. S2). This was likely due to the colder temperatures in the austral winter (Gilbert et al., 2012), with communities from Chile having the lowest phylogenetic diversity coupled with the most southern position. Temperature was recently deemed the best predictor of bacterial diversity in surface waters (Ibarbalz et al., 2019), but we would need to repeat our study during the austral summer and record environmental measurements to verify whether temperature is the best explanation for the pattern we observed.

When examining the beta diversity of the bacterial communities, the ordination technique we used displayed a distinct clustering of samples by incubation location. Statistical testing further confirmed that the incubation location played the largest role in determining microbial community structure. This has previously been observed on a more regional scale along a salinity gradient in the Baltic Sea (Oberbeckmann, Kreikemeyer & Labrenz, 2018), and between open-ocean samples from the Northern Atlantic Gyre and the Northern Pacific Subtropical Gyre (Zettler, Mincer & Amaral-Zettler, 2013; Bryant et al., 2016). In our study, bacterial communities from plastic-coated slides that were incubated at Northern Hemisphere locations (Cape Verde, Japan) had more distinct ASV signatures (29.4% and 32.0%, respectively) than those from the Southern Hemisphere. Although replicates from each location in this study clustered separately, those from the Southern Hemisphere (Chile, South Africa) were more similar to each other than those from the Northern Hemisphere, potentially indicating hemisphere, i.e., seasonal influence, to act as a secondary driver of bacterial community composition, which has been suggested previously on a regional scale (Oberbeckmann et al., 2014). In summary, our study magnifies regional, spatial, and temporal trends on a more global scale.

Implications of incubation location dependencies are important with respect to the identification of potential plastic degrading bacteria. We investigated the presence/absence of certain bacterial groups with respect to hydrocarbon degrading bacteria, mentioned in recent literature (Urbanek, Rymowicz & Mirończuk, 2018; Danso, Chow & Streit, 2019). We found some interesting patterns, but cannot conclude any plastic degrading capabilities from our amplicon data alone. Alcanivorax is part of the obligate hydrocarbonoclastic bacteria (OHCB) group (Cafaro et al., 2013) and is for example differentially abundant in our dataset and almost exclusively found in South Africa and Cape Verde, other members like Ketobacter were absent only from the Chilean sampling location, but on the other hand Oleiphilus was found at all sampling locations. Other bacteria expected to be involved in plastic degradation, such as Erythrobacter (absent from Cape Verde) and Arcobacter (mainly present in Chile) are also present only at some locations (Urbanek, Rymowicz & Mirończuk, 2018), hinting to the Baas Becking hypothesis “everything is everywhere but the environment selects” (1934), as no particular enrichment was apparent.

Influence of plastic type on bacterial community composition

No significant differences in the alpha diversity (Pielou’s evenness, Faith’s PD) of the bacterial communities that established on the different substrate types (HDPE, LDPE, PA, PMMA, glass-slide controls) were detected after five weeks of incubation in the coastal water column. In this study, however, the pooled control replicates exhibited the largest within-group variation in diversity of all substrate types. This could indicate that the set of bacteria that can colonize plastics is less diverse than those that can establish on other and more natural substrates (e.g., driftwood, seaweed, rocks). Additionally, the mean phylogenetic diversity of the communities that established on the HDPE replicates was slightly lower than those that colonized the other substrates. This could have been driven by the general hydrophobicity of the polymer, as PE ranks among the most hydrophobic polymers, which are also the least vulnerable to enzymatic attack (Min, Cuiffi & Mathers, 2020). Furthermore, HDPE is characterized by a higher degree of crystallinity in comparison to LDPE further impeding potential microbial colonization and enzymatic accessibility. The inherent buoyancy of each polymer type had no apparent influence on colonization patterns here. Nonetheless, it needs to be noted that positively buoyant polymers like HDPE and LDPE have a much greater potential for dispersion in the marine environment than negatively buoyant polymers like PA and PMMA, perhaps leading to different colonization patterns if not fixed to one location as done in this study. Pabortsava & Lampitt (2020) identify PE particles as the most abundant microplastic in the upper water layer in the Atlantic Ocean in comparison to polypropylene and polystyrene, supporting its dispersion potential.

Previous research has reached a consensus that the biofilm-forming bacterial communities found on plastics differ significantly from those found free-living in seawater (see review, Amaral-Zettler, Zettler & Mincer, 2020) or on natural substrates, such as wood (Zettler, Mincer & Amaral-Zettler, 2013; Ogonowski et al., 2018). In this study, no significant difference in community composition between the polymer types was observed, a finding that is both corroborated (Oberbeckmann, Kreikemeyer & Labrenz, 2018; Oberbeckmann & Labrenz, 2020; Dudek et al., 2020) and challenged (Lobelle & Cunliffe, 2011; Kirstein et al., 2016; Kirstein et al., 2019) by earlier research. Oberbeckmann et al. (2014) suggested that communities at early time points in the colonization process are more likely to reveal polymer-specificity, while communities that establish on different polymers should gradually converge over time as the biofilms mature (Harrison et al., 2014). Plastic-specific patterns in bacterial community composition have emerged during incubations as short as two minutes (Harrison et al., 2014) and two weeks (Ogonowski et al., 2018), but also after 21 months of incubation when closely attached, mature biofilms were selectively enriched under controlled conditions (Kirstein et al., 2019). Generally, it has been agreed that polymer type plays a minor role in determining bacterial Plastisphere community composition once a mature biofilm has formed, especially when compared to biogeography (Oberbeckmann & Labrenz, 2020). Since in our study the substrates were incubated for five weeks, it could be that any initially existing differences between the polymers and/or between polymers and glass disappeared during the course of biofilm maturation.

Bacterial diversity of the Plastisphere

The data presented here demonstrates that two bacterial phyla, Proteobacteria (classes Alphaproteobacteria and Gammaproteobacteria) and Bacteroidota (class Bacteroidia), dominated the bacterial communities across all substrate types, irrespective of incubation location. A recent meta-analysis, which reanalyzed 16S rRNA gene amplicon data from 35 Plastisphere studies, revealed the successive colonization of the Plastisphere (Wright, Langille & Walker, 2020). The authors found that Alphaproteobacteria are significantly more abundant at early time points of succession, while, at a later stage, a significant increase in Bacteroidia usually coincides with the arrival of Gammaproteobacteria. Previous findings of bacterial Plastisphere communities have a high level of taxonomic overlap with this study (Zettler, Mincer & Amaral-Zettler, 2013; Bryant et al., 2016; Oberbeckmann, Osborn & Duhaime, 2016; Kesy et al., 2019). Many of the taxa that were found to be highly abundant on the marine-incubated plastic-coated slides were also prevalent on plastics that were retrieved from the North Atlantic Gyre (Zettler, Mincer & Amaral-Zettler, 2013) and on those that were incubated for five weeks in the North Sea (Oberbeckmann, Osborn & Duhaime, 2016). Common community members include bacteria that prefer a surface-attached lifestyle, i.e., Flavobacteriaceae (Zheng et al., 2018) and Saprospiraceae (McIlroy & Nielsen, 2014), opportunistic colonizers (Rhodobacteriaceae; Dang & Lovell, 2016), and biofilm formers (Hyphomonodaceae; Abraham & Rohde, 2014), which made up 11.66%, 2.64%, 8.80%, and 1.62% of the median bacterial community composition in this study, respectively. Other recurring members of proteobacterial biofilms on plastics, such as the orders Alteromonadales (Zettler, Mincer & Amaral-Zettler, 2013), Oceanospirillales (Roager & Sonnenschein, 2019), and Cellvibronaceae (Oberbeckmann & Labrenz, 2020; Cheng et al., 2021) represented 2.20%, 2.67%, and 0.60% of the median relative community structure, respectively. Micavibrionaceae (0.54%) was recently described as a PA-specific community member (Di Pippo et al., 2020), while Miao et al. (2019) associated Cyclobacteriaceae (2.51%) with PE and PP. Gammaproteobacterial families with median relative abundances >0.5%, which were detected in our study but were not previously documented as members of bacterial Plastisphere communities, included: Halieaceae (6.04%), Arenicellales (0.93%), and Spongiibacteraceae (0.68%).

Bacteria of the genus Vibrio are ubiquitous in the marine environment (Vezzulli et al., 2012). Concerning plastic surfaces, Vibrio have been described as pathogenic “hitchhikers”, profiting from the abundance of debris available to aid in their dispersal (Zettler, Mincer & Amaral-Zettler, 2013; Kirstein et al., 2016; Debroas, Mone & Ter Halle, 2017), while other Vibrio species have been suggested as promising candidates for the remediation of plastics (Danso, Chow & Streit, 2019). Of the many Vibrio species, 12 are categorized as human pathogens (Kokashvili et al., 2015). In this study, Vibrio accounted for 0.82 ± 0.94% of the bacterial Plastisphere community, although most were not classified to the species level. The classified Vibrio species were assigned to Vibrio sp. 343 and CQB-15, V. gallaecius, and V. breoganii, the latter preferring a vegetarian (microalgal) diet (Corzett et al., 2018). None of the Vibrio in our study are categorized as human pathogens. Our results align with those of Kesy et al. (2019), Oberbeckmann, Kreikemeyer & Labrenz (2018) and Oberbeckmann & Labrenz (2020), who suggest that most Vibrio species represent opportunistic biofilm generalists that favor natural substrates, such as wood, over plastic particles. Microplastics are more and more discussed as potential vectors for microorganisms, especially pathogens, multidrug resistant strains and as vectors for chemical pollutants (Shen et al., 2019). Song and colleagues (2020) incubated particles (HDPE, tyre wear, wood) along a salinity gradient from a river to an offshore island in north west Germany sequentially to reconstruct a potential transport of microorganisms, focusing on multidrug resistant Escherichia coli strains. This study approach could be applied on a wider scale to allow for more general statements on microplastics as vectors, but concluded that there is only a low likelihood for dissemination of multidrug resistant E. coli via plastic particles.

Overall, at the ASV-level, we found no evidence for a global “core” bacterial Plastisphere community. Most previous studies defined “core” microbiomes according to OTUs (Kesy et al., 2019; Oberbeckmann, Kreikemeyer & Labrenz, 2018). Traditional OTU (operational taxonomic units) picking strategies usually cluster reads with less than 3% dissimilarity as one bacterial taxon (one OTU), hence, lose resolution and artificially increase the probability of finding a “core” microbiome. Nevertheless, certain taxonomic groups undoubtedly and consistently recurred on marine-incubated plastics in this study, regardless of the geographic location from which the plastic originated, and this may indicate the presence of a functional core microbiome, although this was not directly tested. Furthermore, when looking at the local scale, we found a number of unique ASVs per polymer type at each location. This hints towards polymer-specific Plastispheres rather than towards a “core” bacterial Plastisphere, which is the same across polymer types and locations (Fig. S3), but the investigated number of samples was too small for a robust statistic testing of this hypothesis.

Conclusions

Our 16S rRNA gene amplicon-based study expands the current knowledge about variability in the composition of bacterial Plastisphere communities by including samples from the previously uncharted Southern Hemisphere. Our findings are consistent with previous reports that there is not a defined “global” Plastisphere community but rather many Plastisphere communities, whose community development and compositions are driven primarily by local and location-specific influences. Although, no significant difference in bacterial community composition was detected between the plastic types used in our study, two bacterial phyla (Proteobacteria and Bacteroidia) dominated the community structure of all replicates, irrespective of incubation location or polymer type.

Supplemental Information

Supplemental Information 1 Environmental parameters and site characteristics of each sampling site

The environmental parameters were collected on 28th June (Cape Verde), 15th June (South Africa), 8th June (Japan) and 29th May (Chile) 2019.

Click here for additional data file.

Supplemental Information 2 ASV table

Click here for additional data file.

Supplemental Information 3 Taxonomy table

Click here for additional data file.

Supplemental Information 4 Taxonomic identification of the 18 ASVs shared across three locations by the NCBI blast algorithm

The blastn suite (blastn. Bethesda (MD): National Library of Medicine (US), National Center for Biotechnology Information; 2004–2021. Available from https://blast.ncbi.nlm.nih.gov/Blast.cgi?PROGRAM=blastn&PAGE_TYPE=BlastSearch&BLAST_SPEC=&LINK_LOC=blasttab&LAST_PAGE=blastp) was employed with the rRNA/ITS database (Sayers et al. 2020).

Click here for additional data file.

Supplemental Information 5 Alpha rarefaction curves for all samples before rarefying

Alpha rarefaction curves displaying the absolute number of ASVs present in each sample for all coastal samples.

Click here for additional data file.

Supplemental Information 6 Influence of hemisphere on alpha diversity

Violin plots depicting (A) Pielou’s evenness (p = 0.018) and (B) Faith’s PD (p = 0.001) of plastic replicates within each hemisphere (N = 29): Northern Hemisphere (n = 14), Southern Hemisphere (n = 15). Significant results are depicted with symbolic number coding, where * and ** represent Wilcoxon p-values ≤ 0.05 and ≤ 0.01, respectively.

Click here for additional data file.

Supplemental Information 7 ASV distribution per plastic type per location

All ASVs unique per polymer type were used, regardless of read count, while ASVs also found in the glass controls were subtracted. The Venn diagrams were drawn with InteractiVenn (Heberle et al., 2015).

Click here for additional data file.

Supplemental Information 8 Qiime2 analysis script

Click here for additional data file.

Biofilm samples for this study were collected by student participants and scientists of the international research and student training program GAME (Global Approach by Modular Experiments). These were Paulo Vasconcelos, Leo Gottschalck and Corrine Almeida (Cape Verde), Abril Sánchez, Jonas Barkhau and Martin Thiel (Chile), Daphné Houiller, Uki Kawata and Masahiro Nakaoka (Japan), Matthew Germizhuisen, Silja Blechschmidt and Maya Pfaff (South Africa). We would also like to thank Ina Clefsen and Andrea Hethke, for their thorough and reliable laboratory assistance. We acknowledge financial support by DFG within the funding program Open Access Publizieren.

Additional Information and Declarations

Competing Interests

Author Contributions

Field Study Permissions

Data Availability

The authors declare there are no competing interests.

Ashley K. Coons performed the experiments, analyzed the data, prepared figures and/or tables, authored or reviewed drafts of the paper, and approved the final draft.

Kathrin Busch analyzed the data, prepared figures and/or tables, authored or reviewed drafts of the paper, and approved the final draft.

Mark Lenz conceived and designed the experiments, performed the experiments, authored or reviewed drafts of the paper, and approved the final draft.

Ute Hentschel conceived and designed the experiments, authored or reviewed drafts of the paper, and approved the final draft.

Erik Borchert conceived and designed the experiments, performed the experiments, prepared figures and/or tables, authored or reviewed drafts of the paper, and approved the final draft.

The following information was supplied relating to field study approvals (i.e., approving body and any reference numbers):

Field experiments were approved by: Ministerio da Agricultura e Ambiente, Cabo Verde; Biodiversity and coastal research department of environmental affairs, South Africa; Office for Mainstreaming Biodiversity, Biodiversity Policy Division, Nature Conservation Bureau, Ministry of the Environment, Japan; Profesional Departamento de Conservación de Especies División de Recursos Naturales y Biodiversidad, Ministerio del Medio Ambiente, Gobierno de Chile.

The following information was supplied regarding data availability:

All raw 16S rRNA gene amplicon reads and the sample metadata and attributes are available in the National Center for Biotechnology Information (NCBI) Sequence Read Archive (SRA): PRJNA720815; BioSample accessions: SAMN18680242–SAMN18680277.

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
