# Peer review of "Biogeography rather than substrate type determines bacterial colonization dynamics of marine plastics"

_PeerJ, doi:10.7717/peerj.12135_

## Round 0.1 · original submission · Minor Revisions

All three reviews are favorable and highlight the quality and importance of the study, and all three reviews offer a number of suggestions to improve the manuscript. I anticipate that you should be able to revise the manuscript according to their comments without undue burden.

I would like to emphasize one comment, in particular, which is that the inclusion of ASV tables and metadata tables in the supplement would be very helpful to your readers.

Reviewer 1 ·

Basic reporting

The paper by Coons and colleagues describes a very interesting dataset obtained from experiments conducted with the same setup and analysis protocol in different oceanic regions and continents, in the framework of the GAME project. All experiments were meant to understand whether microbial community composition is driven by different plastic substrates and/or different geographical regions.
I have no comments on the abstract and introduction. The references are very good and sufficient background is provided. Figures are informative and well represented. The paper and the results well address the hypotheses but I wish it could go a little bit further into the discussion.
Since I can only choose between minor and major revision, I check the box of major revision. However, I wish to clarify to the authors and to the Editor that I suggest a moderate revision of the discussion section and that the manuscript in general reads very well.

Experimental design

I found the paper interesting and novel, advancing our current knowledge of the plastisphere community. The text is well written and the methods section is comprehensive, with a detailed and sound statistical approach. The experiments, analysis and statistical tests as well as results are fully described. The experimental design is robust and I think the comparison of different regions with the same setup is of particular interest.

Validity of the findings

The findings are novel and worth of being disseminated within the scientific community as these new evidences advance our knowledge of the topic and may help future research to focus on specific hypotheses on the bacterial community on plastics.
Conclusions are well stated, however I think the discussion could be expanded to include a broader perspective. I think the authors should dare to provide some new ideas arising from their findings. I would not consider that speculation but rather an advancement of the significance of their findings for processes related to plastics' fate in the marine environment, as well as to marine biogeochemistry.

Additional comments

I have some general comments/questions that I think should be considered in the discussion before publication, to expand the findings and reason some possible phenomena. These comments should not be considered as criticism, but rather that I found the paper very interesting and it has raised many questions and curiosity that I hope the authors can address. Please see below:

-In general, what is the motivation for the choice of plastic types? For example, polystyrene is very abundant and frequently found, but not included. Was there a technical motivation? As of this, the reason for choosing both HDPE/LDPE is the abundancy of these two types of plastic? Do you think that the bacterial community (even if the same) on HDPE/LDPE may affect these materials’ buoyancy differently?

-On buoyancy: clearly the materials you used have different properties. Even if they were all incubated at 1 m depth, I think that some discussion on how the microbial community may affect buoyancy and vertical as well as horizontal distribution of these plastic types, when found as particles, should be added. This would give some insights into the possible reasons behind the biogeographical differences. Maybe plastic properties play a role. The ones that are more widely dispersible can reach different marine compartments/depths than other types of materials. This does not change the results of your experiments, which are very precise, but I wonder whether we would find the four plastic types used, in surface waters, and for five weeks to the same extent to be able to conclude that plastic type does not influence much bacterial community composition. Just some thoughts about it.

-Plastic surface properties may play a major role in the attraction of microbes and on the later possibility of being further broke down in smaller debris. Is it possible that the melting of the plastic polymers onto the glass slides changed somehow hydrophobicity or other properties of the plastics, changing their “reactivity” to marine bacteria or their attractiveness to microbes? It is also possible that the treatment of the plastic coating modified the chances of biofilm-adhesion, thereby affecting embrittlement and microbial degradation of the polymers? How would you compare these surfaces to virgin materials or aged debris that can be both found in the marine environment? Moreover, do you think that melting of polymers may have affected the possible leaching of carbon from plastics thereby influencing microbial adhesion and/or composition? Just some ideas to expand comparison possibilities.

-Why did you choose to deploy the plastic coated slides and control slides at one meter depth? How does this influence the bacterial community?

-Lines 328-329: why? Could you give some examples of natural substrates? However, in lines 350-351 there is a somehow contrasting statement: if in a 5 weeks old biofilm you did not notice significant differences between polymer types and between polymers and control slides, the I guess that on natural substrates it may actually happen the same, that the succession of communities in the 5-weeks biofilm won’t be much diverse, that is, the “substrate core” won’t probably make much difference afterwards. Could you expand this?

-What was the reason for 5 weeks incubation? Comparison with other studies, timing of biofilm formation…just curious.

-Line 387: you discuss about pathogens. Although vibrio in your study was not classified as human pathogen, I was thinking that maybe you could spend a few words on how distinct bacterial communities across biogeographic regions may spread to other environments, possibly carrying along pathogens or threats to environments where they were not present before. This may go together with the horizontal dispersal of plastics. It is true that biogeography played a role in bacterial community composition to answer your research question, but since plastic is floating, it would be a nice addition to discuss on potential “cross-contamination” of the plastisphere community. Do you think there are some species that you would certainly not find in some of your locations of study? Again, just some thoughts on expanding the discussion section.

-Line 401: you discuss about a potential functional core microbiome. What kind of microbiome would you expect, and how could this core microbiome, or certain species that you observed, influence the fate of plastics, like their buoyancy and distribution in the water column? For example, are there certain bacterial species associated to specific organic compounds found on plastic particles that may modify their buoyancy? Or certain species that may be more active in degrading the polymers? I found this paper interesting
(Urbanek et al., 2018 https://doi.org/10.1007/s00253-018-9195-y) and maybe you could add some hypotheses on how distinct biogeography may affect the bacterial communities’ capability of not just adhere on plastic, but to actually modify it?

Reviewer 2 ·

Basic reporting

The language was good and reporting style clear and easy to follow. Overall, the structure, referencing, and background provided were professionally done with only very minor language mistakes. Raw data, however, is largely missing from the supplement. These include: ASV and taxonomy tables, and pairwise distance matrices used for statistical tests. Additionally, for replicability and transparency, all sequences generated within this study must be deposited in an archive and made accessible to the public, which was not indicated here.

Experimental design

The primary aim of the study is an interesting one, and preliminarily addresses differences in microplastic-associated bacterial communities on a more global scale and the factors that shape them. Understandably, conducted as a part of a student training program, there exist quite a few gaps - most notable of which is the lack of recorded environmental data. This, however, is exacerbated by a further lack of more in-depth descriptions of the incubation experiment in itself. Method of deployment and, perhaps more importantly, description of the conditions at the incubation sites are missing completely. What are the influencing water bodies and how are the hydronamic patterns at these sites? These would all greatly supplement the report overall but also the discussion, where inferences made are weakened by the lack of measured environmental data. With the inclusion of some environment descriptions, more meaningful inferences can be made of the data obtained in this study.

Validity of the findings

The results provided are clear and necessarily replicated. Conclusions made, well-linked to the primary aims of the study, were reasonable and provided the right amount of speculation without overstretching assumptions.

Additional comments

1. Line 39 – 40: ‘…estimated that 3 % of annual plastic production enters the ocean each year…’ should be ‘estimated that 3 % of plastic produced annually enter the ocean each year…’

2. Line 52: ‘…describe the microbial life inhabiting marine plastic debris…’. The word ‘inhabit’ by definition means to live in which is not quite the case here. Perhaps a more precise word such as colonize or attached to?

3. Line 54: ‘…into contact with an environment’ should be ‘…into contact with the environment’.

4. Line 69: ‘…duration of biofilm growth’ might more accurately be replaced with ‘age of biofilm’.

5. Line 85: ‘naturally occurring’ should be ‘naturally-occurring’.

6. Line 109: ‘…was melted onto one side of individual glass microscope slides’ should be ‘were individually melted onto one side of glass microscope slides’.

7. Lines 105 – 110: Manufacturer’s information provided for the different polymer types are incomplete – disclosed with some are the company name and location while for others this information is missing.

8. Lines 110 – 113: How were the glass slides deployed? What were they attached to? A brief description of the incubation design might prove helpful here. Additionally, please provide some description of the conditions at the incubation sites. If one of the objectives of the paper is to investigate the role of environmental conditions in shaping biofilm community composition, more information on the characteristics that mark the studied environments are crucial here.

9. Lines 113 – 114: Perhaps this ‘RNA-later-type solution’, if not the actual brand name for a purchased solution but self-prepared, should better be referred to as a stabilization solution or buffer.

10. Line 119: No comma after ‘approved by’.

11. Line 123: Semicolon missing between ‘Japan’ and ‘and’.

12. Line 128: ‘…the surface-scraped biological material…’ should be ‘the biological material scraped off the surface of each slide…’ as the language used here implies that the surface of the biological material was scraped, not that of the slides.

13. Line 144: Open bracket missing before ‘5’-’.

14. Line 150: Please include link to archive where the sequences in this study have been deposited (e.g. Sequence Read Archive (SRA) accession number).

15. Lines 163 – 164: What parameters were used for classify-sklearn? Also, which parameters were used to train the classifier?

16. Line 165: Unassigned sequences on which taxonomic level?

17. Line 167: ‘…was rarefied to 8000 sequences…’ I think a brief justification (like the one from line 225 on subsampling based on lowest read count) should also be included here.

18. Lines 174 – 176: I believe the ‘RCore Team, 2019’ citation should be referenced following mention of the stats package as it is a R-core package while the analyses were performed in RStudio which should be mentioned and also cited differently (i.e. RStudio Team (2020). RStudio: Integrated Development for R. RStudio, PBC, Boston, MA URL http://www.rstudio.com/)

19. Line 178: ‘free, open-source…’ is redundant. Open-source in itself means that the software is freely available.

20. Lines 180 – 181: I am not sure it is correct to refer to Faith’s diversity index as a measure of phylogenetic relatedness. It is correct that the calculation of Faith’s PD is based on the phylogeny of the community but it measures the diversity/richness. Faith’s PD = the sum of the branch lengths that span the tree, hence the greater the sum of the branch lengths, the greater the number of branches. One cannot, however, infer the relatedness of individuals solely based on this metric. For example, a high PD value could stem from a highly branched clade of closely related individuals or from greater separation closer to the root of the tree.

21. Line 183: Not sure there is such a thing as ‘ASV-level alpha diversity’. The violin plots serve to visualize the distribution of observations (among the different samples of each polymer type). ASVs are proxies for species, so what is ASV-level alpha diversity?

22. Lines 184 – 186: Why were the controls included in the experimental design just to be removed from the comparison within the analysis? It is unclear here from which analyses exactly the controls were excluded seeing as how, based on the figures/tables provided, they were included in the alpha diversity calculations, ordinations, and selected multivariate analyses.

23. Line 189: Would be better to mention the significance level as α = 0.05. Same goes for line 201.

24. Line 188 and line 191: The use of the word ‘sample types’ implies or could be understood as a comparison just between the different substrate types. Perhaps better to just refer to it as ‘samples’ here which would then encompass the different incubation sites and substrate types.

25. Line 193: ‘…assistance of the phyloseq R package’ should be ‘assistance of the R package phyloseq’.

26. Lines 195 – 203: What was the motivation to calculate phylogenetic distances based only on unweighted metrics and not weighted? Also, ‘UniFraq’ should be ‘UniFrac’.

27. Lines 201 – 203: PERMANOVA usually needs to be accompanied by PERMDISP (Permutational Analysis of Multivariate Dispersions) which detects for large dispersions/variations within groups and more importantly ensures that significant PERMANOVA p-values are really the product of between-group differences and not within-group variation.

28. Lines 205 – 206 (and again in line 207): What is ‘median relative community composition’?
Additionally, consider rephrasing this sentence as it is presented more as a result (particularly: ‘A bubble plot displayed phyla…’) rather than an action that was performed.

29. Lines 208 – 210: ‘Family names’ and ‘order names’ may be better referred to as assignments (e.g. taxonomic assignments on the family and order levels). Also, a little confused by this sentence. Were the reads removed or just collapsed to the next highest classified taxonomic level?

30. Lines 221 – 222: ‘…36 samples from four polymer types’ should be ’36 samples composed of/consisting of four polymer types’. Also, if there were 2 slides of each substrate type at each site but you ended up with 36 samples in total, what happened to the other 4 slides?

31. Line 228: ‘…reads were distributed among 12361 unique features’ should be ‘reads were made up of 12361 unique features’.

32. Line 231: Might be helpful to start here with a brief mention of the mean or range of values observed for the different diversity indices before getting into the comparisons. These are important to provided a clearer idea of the diversity of the communities found on the different substrates at each site.

33. Line 234 – 235: For Pielou’s evenness ‘p < 0.001’ but for Faith’s PD ‘p = 0.003’? Why is one a range and the other not?

34. Table 1B: ‘permutation’ should be ‘permutations’.

35. Lines 263 – 264: Again, how does inclusion of the controls interfere with any of the analyses? Perhaps some confusion here but ‘removed from analysis’ encompasses all analyses. Was it the case that the control samples were excluded from ALL analyses, including statistical tests? This is important and if not the intended meaning, please consider rephrasing.

36. Lines 300 – 303: ‘Communities on plastic-coated glass slides from all locations were similar in evenness, while the communities from the equator (Cape Verde) had a significantly higher evenness than the mid-latitude incubation locations (Chile, Japan, South Africa)’. This sentence contradicts itself as it is easily understood as ‘Communities from all locations were similar yet significantly different’. Does the first part of the sentence refer to a comparison between substrate types? If so, some rephrasing is required because that is not clear.

37. Lines 333 – 336: How does the wide range of diversity values observed among control samples indicate that it is has a greater diversity than those of plastics? Does a wide distribution of data points not indicate only that there is, as stated by the authors, variation within the group?

38. Lines 355 – 357: Seems a little contradictory that in the previous statement you mention that there is currently no clear consensus on whether bacteria exhibit substrate-specificity yet in this sentence it is ‘generally […] agreed’ that they do not. It then becomes quite convoluted in the following sentence, where authors attribute the lack of specificity observed in this study, to the age of the biofilm given especially that it was just mentioned that other studies have observed such specificity even after longer periods (i.e. 21 months: Kirstein et al. 2019).

39. Lines 385 – 387: Were these families also detected in this study and associated with these polymer types or why is this being mentioned here?

40. Line 390: Cellvibrionaceae is referred to here as ‘detected in our study but were not previously documented as members of bacterial Plastisphere communities’ but was already mentioned in line 383 to be reported by Oberbeckmann & Labrenz and Cheng et al..

41. Lines 401 – 404: How did your findings align with those of Kesy et al. and Oberbeckmann et al.? Were Vibrio found more abundantly on the glass controls used in this study or is this claim made purely based on the fact that Vibrio made up a very small proportion of the overall community? Bear in mind that the latter means a very different thing from the former.

Reviewer 3 ·

Basic reporting

This was a clear and well-written paper that I enjoyed reviewing. The authors have provided a good overview of current plastisphere research, however, there are a handful of recent papers that do characterise the plastisphere in the Southern hemisphere that I think should be discussed/mentioned (there are also potentially others published during 2021, but I most recently searched the literature at the beginning of this year):
1. Cornejo-D'Ottone et al. 2020 (16S rRNA gene sequenced – focus of the paper on greenhouse gas cycling, sampling in subtropical South Pacific gyre) https://doi.org/10.1016/j.chemosphere.2019.125709
2. Lacerda et al. 2020 (focused on fungi not prokaryotes – sequenced ITS2 and 18S rRNA genes, Western South Atlantic and Antarctic) https://doi.org/10.1111/mec.15444
3. Zaiko et al. 2016 (focused on eukaryotes – 18S rRNA gene sequenced, incubations in Lyttelton, New Zealand) https://doi.org/10.1080/08927014.2016.1186165
4. Tobias-Hunefeldt et al. 2020 (16S and 18S rRNA genes sequenced, incubations in Otago, New Zealand) https://doi.org/10.1038/s41396-020-00833-6
5. Lee et al. 2016 (16S rRNA gene 454 pyrosequencing, King George Island, Antarctica) http://dx.doi.org/10.7845/kjm.2016.6005
6. Krause et al. 2020 (16S rRNA gene sequenced, Peru basin, South-East Equatorial Pacific – potentially 2 decades on abyssal seafloor) https://doi.org/10.1038/s41598-020-66361-7
While I agree that some of these papers have a different focus than the current study, I think that the findings of this study should be put into context with these papers.
I also appreciate that the authors have uploaded their raw sequencing data to the SRA, although I think it would be beneficial to also provide the processed ASV tables (with taxonomy) as well as metadata (e.g., any information on location, physicochemical parameters at the time of sample collection, etc.) as supplementary information. Ideally, any scripts used for analysis should also either be provided or a link to this on Github added.

Experimental design

See above related to discussing the findings of other Southern hemisphere plastisphere studies. This research is, however, still relevant and I think the experimental design has been carefully designed. The only thing I found slightly disappointing was to only have two replicates for each substrate type within each location – including at least three would have enabled further statistical testing. It is obviously not possible to include this retrospectively, but I hope the authors take this into account for future studies. This would have enabled more comparison within locations, which are where the largest differences in plastisphere composition between substrate types are thought to occur.
There are some further specific questions that I had below.

Validity of the findings

Overall I think that the authors have been careful in not overstating their results, however, they say in the abstract that the “study sheds light on the question of whether bacterial communities on plastic debris are shaped by the physicochemical properties of the substrate they grow on or by the marine environment in which the plastics are immersed”. Whilst it’s clear that overall the largest differences in community composition are due to location/other parameters related to this, they weren’t able to look for differences in community composition between the different substrates within the locations that they study and I therefore think that this as well as any other similar statements should be revised accordingly. I also think it would be interesting to pool the two PE substrates and the other two substrates in order to look for differences – on the NMDS plots for some locations it does look like there is slight separate grouping and I would be interested to see this investigated further. It would also be good to show the NMDS plot with weighted unifrac as well as the unweighted distance that is shown.
The authors do discuss this briefly (line 301), but I would have liked to see further discussion of the underlying differences between each sample location, e.g. temperature, salinity, light availability, given that the sampling was at different seasons for Northern/Southern hemisphere.

Additional comments

This was a nice, well-written paper that I think contributes nicely to our understanding of the plastisphere. Further to the above comments, I had a number of specific queries/comments:
Line 84 and others: note that the study referenced doesn’t use chitin. While I agree that this is a naturally occurring substrate, wood is usually included because it is assumed to be a relatively inert substrate in the ocean, whereas chitin is much more rapidly degraded.
Throughout: Note that Unifrac should be spelt with a ‘c’ and not a ‘q’.
Line 154: It’s not clear to me why the authors used the single-end DADA2 method when they had paired end data – were the reads joined prior to this? Please clarify.
Line 164: Was the tree de novo? Did the authors verify the validity of the tree? In my experience, amplicon sequences do not make robust trees and these usually do not reflect the generally accepted phylogeny. I would have preferred to see this step done via insertion into an existing tree (e.g. the SEPP method within QIIME2), particularly as this tree was presumably used for the beta diversity analyses.
Lines 182-187: at which phylogenetic level were these tests performed? And I assume that these are for differential abundance testing? There are many microbiome-specific methods (ANCOM, ALDeX, Corncob…) that I think would have been more appropriate here; while not completely necessary, I do think that more information needs to be provided here.
Line 191: Was this absolute count before rarefying? Related, the rarefaction curces in Fig. S1 should ideally show all samples, and should be before rarefying.
Line 252-257: What is meant by diversity? The number of ASVs? Relative abundance? Ranges for each of these eleven phyla would also be useful to add here.
Lines 264-265: Add references
Line 280: Does NCBI BLAST give any further indication to which taxa these are?
Fig. 2 legend: Replace “insignificant” with “non-significant”
Fig. 5: I think this figure would benefit from an additional colour scheme representing abundance or an additional column indicating the maximum relative abundance for each phylum as it is currently a bit difficult to discern the differences in abundance. Also, for the lower abundance phyla, are these each present on each replicate within the substrates? Something to indicate whether this is the case or not should be added.
Fig. 6: Are the “slices” shown here proportional to abundance? A more detailed description of this chart as well as an idea of the number of ASVs that each family encompasses should be added. I think showing either the family or genera as something like a heatmap showing the differences between substrates would also be useful – having this only at the phylum (i.e. Fig. 5) is a bit high to really have a good idea of the differences/similarities here.

---

## Round 0.2 · Minor Revisions

The revised manuscript has addressed nearly all of the reviewers' concerns and requires only a few minor additional revisions. Reviewer 3 makes several valid points, some of which can be considered differences of opinion, and for which we can "agree to disagree". For example, I share the reviewer's concern about constructing trees do novo for diversity estimates, but this is a fairly common technique, and the methodology is adequately described in the manuscript. However, I agree with the reviewer that including the justification for single-end DADA2 analysis is a good idea, since this is not standard practice and should be highlighted. In addition, I agree that the plot in Figure 6 would benefit from some indication of the total number of ASVs represented in the plot, so that the reader can appreciate the scale of the relative differences. (In other words, is this plot showing 100 ASVs and 1,000 sequence counts or 100,000 ASVs and a million sequence counts?) I anticipate that making these last few edits should not be overly burdensome.

Reviewer 1 ·

Basic reporting

Dear Authors,

thanks for your rebuttal letter and for addressing the comments from both reviewers in your revised version. I found your manuscript very interesting and I do not have further comments.

Experimental design

I appreciated the inclusion of raw data and codes as suggested by the reviewer as well as supplementary data indicating sampling locations and environmental parameters.

Validity of the findings

I appreciate the authors' effort to address my previous comments and questions and I think now with the additions the manuscript has much improved.

Reviewer 3 ·

Basic reporting

No comment

Experimental design

No comment

Validity of the findings

No comment

Additional comments

I am pleased with the revisions made and think that most of my concerns as well as those of the other reviewers are now covered. I have only a few minor comments remaining:
- It was disappointing that the authors didn't try to include a combined analysis of the substrates (i.e. HDPE+LDPE and PA+PMMA), or comment on their choice not to do this.
- I am fine with the authors' reasoning to use only the single-ended sequencing data, but think this should be added to the text as (presumably) the data uploaded to the SRA is double-ended. If future researchers attempt to reuse this data then this may be relevant.
- I agree that their tree was probably fine, but maybe I didn't make my point/concern clear enough. In my past experience, creating a de novo tree with amplicon sequences (i.e. reads that cover only a small fraction of the 16S rRNA gene) has led to a tree where the majority of placements are fine, but there will often be something strange such as the Gammaproteobacteria being split into two groups, separated by the Alphaproteobacteria. I was therefore mainly getting at whether - as they do not provide the tree that they used, or a visualisation of it - they verified that it was actually correct according to the accepted phylogeny before using it for the unifrac analyses. SEPP is one method that I gave an example of, although not what the authors necessarily have to use - my only point was that if there were issues with the tree due to the short amplicon sequences (and I would be surprised if there were none) then insertion into a larger tree would alleviate these. Hopefully I have clarified my comment sufficiently now.
- "Insignificant" should also be switched to "non-significant" in Fig. 3.
- For Fig. 6 the authors explained this more thoroughly to me, but this also needs to be added to the figure legend. I have not previously come across such a plot and believe this may be the case for many others too. Explicitly stating exactly what this is and that e.g. that the size of a chuck is proportional to median relative abundance would be useful. I disagree that adding ASV counts would be misleading, as it would give readers an idea of the diversity within different groups (although they could choose to do this in another way - I only gave a suggestion) - there is currently not anything that illustrates whether e.g. all of the relative abundance accounted for by Bacteroidota members is due to only 3 ASVs while the relative abundance accounted for by Gammaproteobacteria is accounted for by 100 ASVs. This is clearly an extreme example but I do think it would be interesting to have something that gives the reader a better indication of the composition at lower taxonomic levels, whatever the authors prefer this to be. It seems odd to have nothing that gives the reader an idea of how the composition differs at very fine resolutions, particularly as they discuss things like the relative abundance of Vibrio spp. but give the reader no opportunity to visualise this, and in fact the first mention of this is in the discussion.

---

## Round 0.3 · accepted · Accept

Thank you for your thoughtful responses to all reviewer concerns and comments. I look forward to seeing this work published.